# CBOW Is Not All You Need: Combining CBOW with the Compositional Matrix Space Model

**Florian Mai**
Idiap Research Institute
Martigny, Switzerland
`florian.mai@idiap.ch`

**Lukas Galke**
Kiel University / ZBW
Germany
`lga@informatik.uni-kiel.de`

**Ansgar Scherp**
University of Essex
United Kingdom
`ansgar.scherp@essex.ac.uk`

## Abstract

Continuous Bag of Words (CBOW) is a powerful text embedding method. Due to its strong capabilities to encode word content, CBOW embeddings perform well on a wide range of downstream tasks while being efficient to compute. However, CBOW is not capable of capturing the word order. The reason is that the computation of CBOW's word embeddings is commutative, i.e., embeddings of XYZ and ZYX are the same. In order to address this shortcoming, we propose a learning algorithm for the Continuous Matrix Space Model (Rudolph & Giesbrecht, 2010), which we call Continual Multiplication of Words (CMOW). Our algorithm is an adaptation of word2vec (Mikolov et al., 2013a), so that it can be trained on large quantities of unlabeled text. We empirically show that CMOW better captures linguistic properties, but it is inferior to CBOW in memorizing word content. Motivated by these findings, we propose a hybrid model that combines the strengths of CBOW and CMOW. Our results show that the hybrid CBOW-CMOW-model retains CBOW's strong ability to memorize word content while at the same time substantially improving its ability to encode other linguistic information by 8%. As a result, the hybrid also performs better on 8 out of 11 supervised downstream tasks with an average improvement of 1.2%.

## 1 Introduction

Word embeddings are perceived as one of the most impactful contributions from unsupervised representation learning to natural language processing from the past few years (Goth, 2016). Word embeddings are learned once on a large-scale stream of words. A key benefit is that these pre-computed vectors can be re-used almost universally in many different downstream applications. Recently, there has been increasing interest in learning universal *sentence* embeddings. Perone et al. (2018) have shown that the best encoding architectures are based on recurrent neural networks (RNNs) (Conneau et al., 2017; Peters et al., 2018) or the Transformer architecture (Cer et al., 2018). These techniques are, however, substantially more expensive to train and apply than word embeddings (Hill et al., 2016; Cer et al., 2018). Their usefulness is therefore limited when fast processing of large volumes of data is critical.

More efficient encoding techniques are typically based on aggregated word embeddings such as Continuous Bag of Words (CBOW), which is a mere summation of the word vectors (Mikolov et al., 2013a). Despite CBOW's simplicity, it attains strong results on many downstream tasks. Using sophisticated weighting schemes, the performance of aggregated word embeddings can be further increased (Arora et al., 2017), coming even close to strong LSTM baselines (Rücklé et al., 2018; Henao et al., 2018) such as InferSent (Conneau et al., 2017). This raises the question how much benefit recurrent encoders actually provide over simple word embedding based methods (Wieting & Kiela, 2019). In their analysis, Henao et al. (2018) suggest that the main difference may be the

ability to encode word order. In this paper, we propose an intuitive method to enhance aggregated word embeddings by word order awareness.

The major drawback of these CBOW-like approaches is that they are solely based on addition. However, *addition is not all you need*. Since it is a commutative operation, the aforementioned methods are not able to capture any notion of word order. However, word order information is crucial for some tasks, e.g., sentiment analysis (Henao et al., 2018). For instance, the following two sentences yield the exact same embedding in an addition-based word embedding aggregation technique: "The movie was not awful, it was rather great." and "The movie was not great, it was rather awful." A classifier based on the CBOW embedding of these sentences would inevitably fail to distinguish the two different meanings (Goldberg, 2017, p. 151).

To alleviate this drawback, Rudolph & Giesbrecht (2010) propose to model each word as a matrix rather than a vector, and compose multiple word embeddings *via matrix multiplication rather than addition*. This so-called *Compositional Matrix Space Model* (CMSM) of language has powerful theoretical properties that subsume properties from vector-based models and symbolic approaches. The most obvious advantage is the non-commutativity of matrix multiplication as opposed to addition, which results in order-aware encodings.

In contrast to vector-based word embeddings, there is so far no solution to effectively train the parameters of word matrices on large-scale *unlabeled* data. Training schemes from previous work were specifically designed for sentiment analysis (Yessenalina & Cardie, 2011; Asaadi & Rudolph, 2017). Those require complex, multi-stage initialization, which indicates the difficulty of training CMSMs. We show that CMSMs can be trained in a similar way as the well-known CBOW model of word2vec (Mikolov et al., 2013a). We make two simple yet critical changes to the initialization strategy and training objective of CBOW. Hence, we present the first unsupervised training scheme for CMSMs, which we call *Continual Multiplication Of Words* (CMOW).

We evaluate our model's capability to capture linguistic properties in the encoded text. We find that CMOW and CBOW have properties that are complementary. On the one hand, CBOW yields much stronger results at the word content memorization task. CMOW, on the other hand, offers an advantage in all other linguistic probing tasks, often by a wide margin. Thus, we propose a hybrid model to jointly learn the word vectors of CBOW and the word matrices for CMOW.

Our experimental results confirm the effectiveness of our hybrid CBOW-CMOW approach. At comparable embedding size, CBOW-CMOW retains CBOW's ability to memorize word content while at the same time improves the performance on the linguistic probing tasks by 8%. CBOW-CMOW outperforms CBOW at 8 out of 11 supervised downstream tasks scoring only 0.6% lower on the tasks where CBOW is slightly better. On average, the hybrid model improves the performance over CBOW by 1.2% on supervised downstream tasks, and by 0.5% on the unsupervised tasks.

In summary, our contributions are: (1) For the first time, we present an unsupervised, efficient training scheme for the Compositional Matrix Space Model. Key elements of our scheme are an initialization strategy and training objective that are specifically designed for training CMSMs. (2) We quantitatively demonstrate that the strengths of the resulting embedding model are complementary to classical CBOW embeddings. (3) We successfully combine both approaches into a hybrid model that is superior to its individual parts.

After giving a brief overview of the related work, we formally introduce CBOW, CMOW, and the hybrid model in Section 3. We describe our experimental setup and present the results in Section 4. The results are discussed in Section 5, before we conclude.

## 2 RELATED WORK

We present an algorithm for learning the weights of the Compositional Matrix Space Model (Rudolph & Giesbrecht, 2010). To the best of our knowledge, only Yessenalina & Cardie (2011) and Asaadi & Rudolph (2017) have addressed this. They present complex, multi-level initialization strategies to achieve reasonable results. Both papers train and evaluate their model on sentiment analysis datasets only, but they do not evaluate their CMSM as a general-purpose sentence encoder.

Other works have represented words as matrices as well, but unlike our work not within the framework of the CMSM. Grefenstette & Sadrzadeh (2011) represent only relational words as matrices. Socher et al. (2012) and Chung & Bowman (2018) argue that while CMSMs are arguably more expressive than embeddings located in a vector space, the associativeness of matrix multiplication does not reflect the hierarchical structure of language. Instead, they represent the word sequence as a tree structure. Socher et al. (2012) directly represent each word as a matrix (and a vector) in a recursive neural network. Chung & Bowman (2018) present a two-layer architecture. In the first layer, pre-trained word embeddings are mapped to their matrix representation. In the second layer, a non-linear function composes the constituents.

Sentence embeddings have recently become an active field of research. A desirable property of the embeddings is that the encoded knowledge is useful in a variety of high-level downstream tasks. To this end, Conneau & Kiela (2018) and Conneau et al. (2018) introduced an evaluation framework for sentence encoders that tests both their performance on downstream tasks as well as their ability to capture linguistic properties. Most works focus on either i) the *ability* of encoders to capture appropriate semantics or on ii) training objectives that give the encoders *incentive* to capture those semantics. Regarding the former, large RNNs are by far the most popular (Conneau et al., 2017; Kiros et al., 2015; Tang et al., 2017; Nie et al., 2017; Hill et al., 2016; McCann et al., 2017; Peters et al., 2018; Logeswaran & Lee, 2018), followed by convolutional neural networks (Gan et al., 2017). A third group are efficient methods that aggregate word embeddings (Wieting et al., 2016; Arora et al., 2017; Pagliardini et al., 2018; Rücklé et al., 2018). Most of the methods in the latter group are word order agnostic. Sent2Vec (Pagliardini et al., 2018) is an exception in the sense that they also incorporate bigrams. Despite also employing an objective similar to CBOW, their work is very different to ours in that they still use addition as composition function. Regarding the training objectives, there is an ongoing debate whether language modeling (Peters et al., 2018; Ruder & Howard, 2018), machine translation (McCann et al., 2017), natural language inference (Conneau et al., 2017), paraphrase identification (Wieting et al., 2016), or a mix of many tasks (Subramanian et al., 2018) is most appropriate for incentivizing the models to learn important aspects of language. In our study, we focus on adapting the well-known objective from word2vec (Mikolov et al., 2013a) for the CMSM.

## 3 METHODS: CBOW AND CMOW

We formally present CBOW and CMOW encoders in a unified framework. Subsequently, we discuss the training objective, the initialization strategy, and the hybrid model.

### 3.1 TEXT ENCODING

We start with a lookup table for the word matrices, i.e., an embedding, $\boldsymbol{E} \in \mathbb{R}^{m \times d \times d}$, where $m$ is the vocabulary size and $d$ is the dimensionality of the (square) matrices. We denote a specific word matrix of the embedding by $\boldsymbol{E}[\cdot]$. By $\Delta \in \{\sum, \prod\}$ we denote the function that aggregates word embeddings into a sentence embedding. Formally, given a sequence $s$ of arbitrary length $n$, the sequence is encoded as $\Delta_{i=1}^{n} \boldsymbol{E}[s_i]$. For $\Delta = \sum$, the model becomes CBOW. By setting $\Delta = \prod$ (matrix multiplication), we obtain CMOW. Because the result of the aggregation for any prefix of the sequence is again a square matrix of shape $d \times d$ irrespective of the aggregation function, the model is well defined for any non-zero sequence length. Thus, it can serve as a general-purpose text encoder.

Throughout the remainder of this paper, we denote the encoding step by $\mathrm{enc}_{\Delta}^{\boldsymbol{E}}(s) := \mathrm{flatten}\left(\Delta_{i=1}^{n} \boldsymbol{E}[s_i]\right)$, where flatten concatenates the columns of the matrices to obtain a vector that can be passed to the next layer.

### 3.2 TRAINING OBJECTIVE

Motivated by its success, we employ a similar training objective as word2vec (Mikolov et al., 2013b). The objective consists of maximizing the conditional probability of a word $w_O$ in a certain context $s$: $p(w_O \mid s)$. For a word $w_t$ at position $t$ within a sentence, we consider the window of tokens $(w_{t-c}, \ldots, w_{t+c})$ around that word. From that window, a target word $w_O := \{w_{t+i}\}, i \in \{-c, \ldots, +c\}$ is selected. The remaining $2c$ words in the window are used

as the context $s$. The training itself is conducted via negative sampling NEG-k, which is an efficient approximation of the softmax (Mikolov et al., 2013b). For each positive example, $k$ negative examples (noise words) are drawn from some noise distribution $P_n(w)$. The goal is to distinguish the target word $w_O$ from the randomly sampled noise words. Given the encoded input words $\text{enc}_\Delta(s)$, a logistic regression with weights $v \in \mathbb{R}^{m \times d^2}$ is conducted to predict 1 for context words and 0 for noise words. The negative sampling training objective becomes:

$$\log \sigma \left( v_{w_O}^T \, \text{enc}_\Delta^{\boldsymbol{E}}(s) \right) + \sum_{i=1}^{k} \mathbb{E}_{w_i \sim P_n(w)} \left[ \log \sigma \left( -v_{w_i}^T \, \text{enc}_\Delta^{\boldsymbol{E}}(s) \right) \right] \tag{1}$$

In the original word2vec (Mikolov et al., 2013a), the center word $w_O := w_t$ is used as the target word. In our experiments, however, this objective did not yield to satisfactory results. We hypothesize that this objective is too easy to solve for a word order-aware text encoder, which diminishes incentive for the encoder to capture semantic information at the sentence level. Instead, we propose to select a random output word $w_O \sim \mathcal{U}(\{w_{t-c}, \dots, w_{t+c}\})$ from the window. The rationale is the following: By removing the information at which position the word was removed from the window, the model is forced to build a semantically rich representation of the *whole* sentence. For CMOW, modifying the objective leads to a large improvement on downstream tasks by 20.8% on average, while it does not make a difference for CBOW. We present details in the appendix (Section B.1).

## 3.3 INITIALIZATION

So far, only Yessenalina & Cardie (2011) and Asaadi & Rudolph (2017) have proposed algorithms for learning the parameters for the matrices in CMSMs. Both works devote particular attention to the initialization, noting that a standard initialization randomly sampled from $\mathcal{N}(0, 0.1)$ does not work well due to the optimization problem being non-convex. To alleviate this, the authors of both papers propose rather complicated initialization strategies based on a bag-of-words solution (Yessenalina & Cardie, 2011) or incremental training, starting with two word phrases (Asaadi & Rudolph, 2017). We instead propose an effective yet simple strategy, in which the embedding matrices are initialized close to the identity matrix.

We argue that modern optimizers based on stochastic gradient descent have proven to find good solutions to optimization problems even when those are non-convex as in optimizing the weights of deep neural networks. CMOW is essentially a deep linear neural network with flexible layers, where each layer corresponds to a word in the sentence. The output of the final layer is then used as an embedding for the sentence. A subsequent classifier may expect that all embeddings come from the same distribution. We argue that initializing the weights randomly from $\mathcal{N}(0, 0.1)$ or any other distribution that has most of its mass around zero is problematic in such a setting. This includes the Glorot initialization (Glorot & Bengio, 2010), which was designed to alleviate the problem of vanishing gradients. Figure 1 illustrates the problem: With each multiplication, the values in the embedding become smaller (by about one order of magnitude). This leads to the undesirable effect that short sentences have a drastically different representation than larger ones, and that the embedding values vanish for long sequences.

To prevent this problem of vanishing values, we propose an initialization strategy, where each word embedding matrix $\boldsymbol{E}[w] \in \mathbb{R}^{d \times d}$ is initialized as a random deviation from the identity matrix:

$$\boldsymbol{E}[w] := \begin{pmatrix} \mathcal{N}(0, 0.1) & \dots & \mathcal{N}(0, 0.1) \\ \vdots & \ddots & \vdots \\ \mathcal{N}(0, 0.1) & \dots & \mathcal{N}(0, 0.1) \end{pmatrix} + \boldsymbol{I}_d,$$

It is intuitive and also easy to prove that the expected value of the multiplication of any number of such word embedding matrices is again the identity matrix (see Appendix A). Figure 1 shows how our initialization strategy is able to prevent vanishing values. For training CMSMs, we observe a substantial improvement over Glorot initialization of 2.8% on average. We present details in Section B.2 of the appendix.

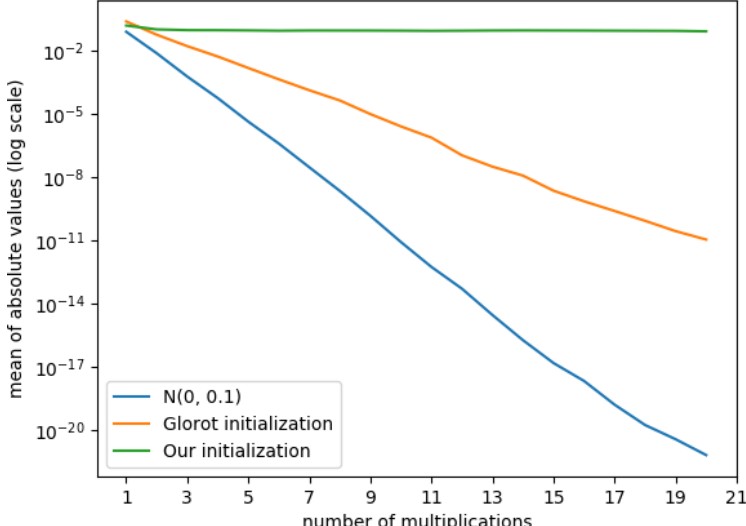

Figure 1: Mean of the absolute values of the text embeddings (y-axis) plotted depending on the number of multiplications (x-axis) for the three initialization strategies. As one can see, the absolute value of the embeddings sharply decreases for the initialization strategies Glorot and $\mathcal{N}(0, 0.1)$ the more multiplications are performed. In contrast, when our initialization method is applied, the absolute values of the embeddings have the same magnitude regardless of the sentence length.

### 3.4 HYBRID CBOW-CMOW MODEL

Due to their different nature, CBOW and CMOW also capture different linguistic features from the text. It is therefore intuitive to expect that a hybrid model that combines the features of their constituent models also improves the performance on downstream tasks.

The simplest combination is to train CBOW and CMOW separately and concatenate the resulting sentence embeddings at test time. However, we did not find this approach to work well in preliminary experiments. We conjecture that there is still a considerable overlap in the features learned by each model, which hinders better performance on downstream tasks. To prevent redundancy in the learned features, we expose CBOW and CMOW to a shared learning signal by training them jointly. To this end, we modify Equation 1 as follows:

$$
\log \sigma \left( v_{w_O}^T [\text{enc}_{\sum}^{\boldsymbol{E_1}}(s); \text{enc}_{\prod}^{\boldsymbol{E_2}}(s)] \right)
$$
$$
+ \sum_{i=1}^{k} \mathbb{E}_{w_i \sim P_n(w)} \left[ \log \sigma \left( -v_{w_i}^T [\text{enc}_{\sum}^{\boldsymbol{E_1}}(s); \text{enc}_{\prod}^{\boldsymbol{E_2}}(s)] \right) \right].
$$

Intuitively, the model uses logistic regression to predict the missing word from the concatenation of CBOW and CMOW embeddings. Again, $\boldsymbol{E_i} \in \mathbb{R}^{m \times d_i \times d_i}$ are separate word lookup tables for CBOW and CMOW, respectively, and $v \in \mathbb{R}^{m \times (d_1^2 + d_2^2)}$ are the weights of the logistic regression.

## 4 EXPERIMENTS

We conducted experiments to evaluate the effect of using our proposed models for training CMSMs. In this section, we describe the experimental setup and present the results on linguistic probing as well as downstream tasks.

### 4.1 EXPERIMENTAL SETUP

In order to limit the total batch size and to avoid expensive tokenization steps as much as possible, we created each batch in the following way: 1,024 sentences from the corpus are selected at random.

After tokenizing each sentence, we randomly select (without replacement) at maximum 30 words from the sentence to function as center words for a context window of size $c = 5$, i.e., we generate up to 30 training samples per sentence. By padding with copies of the neutral element, we also include words as center words for which there are not enough words in the left or the right context. For CBOW, the neutral element is the zero matrix. For CMOW, the neutral element is the identity matrix.

We trained our models on the unlabeled UMBC news corpus (Han et al., 2013), which consists of about 134 million sentences and 3 billion tokens. Each sentence has 24.8 words on average with a standard deviation of 14.6. Since we only draw 30 samples per sentence to limit the batch size, not all possible training examples are used in an epoch, which may result in slightly worse generalization if the model is trained for a fixed number of epochs. We therefore use 0.1% of the 134 million sentences for validation. After 1,000 updates (i.e., approximately every millionth training sample) the validation loss is calculated, and training terminates after 10 consecutive validations of no improvement. Following Mikolov et al. (2013b), we limit the vocabulary to the 30,000 most-frequent words for comparing our different methods and their variants. Out-of-vocabulary words are discarded. The optimization is carried out by Adam (Kingma & Ba, 2015) with an initial learning rate of 0.0003 and $k = 20$ negative samples as suggested by Mikolov et al. (2013b) for rather small datasets. For the noise distribution $P_n(w)$ we again follow Mikolov et al. (2013b) and use $\mathcal{U}(w)^{3/4}/Z$, where $Z$ is the partition function to normalize the distribution.

We have trained five different models: CBOW and CMOW with $d = 20$ and $d = 28$, which lead to 400-dimensional and 784-dimensional word embeddings, respectively. We also trained the Hybrid CBOW-CMOW model with $d = 20$ for each component, so that the total model has 800 parameters per word in the lookup tables. We report the results of two more models: H-CBOW is the 400-dimensional CBOW component trained in Hybrid and H-CMOW is the respective CMOW component. Below, we compare the 800-dimensional Hybrid method to the 784-dimensional CBOW and CMOW models.

After training, only the encoder of the model $\mathrm{enc}_\Delta^E$ is retained. We assess the capability to encode linguistic properties by evaluating on 10 linguistic probing tasks (Conneau et al., 2018). In particular, the Word Content (WC) task tests the ability to memorize exact words in the sentence. Bigram Shift (BShift) analyzes the encoder's sensitivity to word order. The downstream performance is evaluated on 10 supervised and 6 unsupervised tasks from the SentEval framework (Conneau & Kiela, 2018). We use the standard evaluation configuration, where a logistic regression classifier is trained on top of the embeddings.

## 4.2 RESULTS ON LINGUISTIC PROBING TASKS

Considering the linguistic probing tasks (see Table 1), CBOW and CMOW show complementary results. While CBOW yields the highest performance at word content memorization, CMOW out-performs CBOW at all other tasks. Most improvements vary between 1-3 percentage points. The difference is approximately 8 points for CoordInv and Length, and even 21 points for BShift.

The hybrid model yields scores close to or even above the better model of the two on all tasks. In terms of relative numbers, the hybrid model improves upon CBOW in all probing tasks but WC and SOMO. The relative improvement averaged over all tasks is 8%. Compared to CMOW, the hybrid model shows rather small differences. The largest loss is by 4% on the CoordInv task. However, due to the large gain in WC (20.9%), the overall average gain is still 1.6%.

We now compare the jointly trained H-CMOW and H-CBOW with their separately trained 400-dimensional counterparts. We observe that CMOW loses most of its ability to memorize word content, while CBOW shows a slight gain. On the other side, H-CMOW shows, among others, improvements at BShift.

## 4.3 RESULTS ON DOWNSTREAM TASKS

Table 2 shows the scores from the supervised downstream tasks. Comparing the 784-dimensional models, again, CBOW and CMOW seem to complement each other. This time, however, CBOW has the upperhand, matching or outperforming CMOW on all supervised downstream tasks except

Table 1: Scores on the probing tasks attained by our models. Rows starting with "Cmp." show the relative change with respect to Hybrid.

| Dim | Method | Depth | BShift | SubjNum | Tense | CoordInv | Length | ObjNum | TopConst | SOMO | WC |
|-----|--------|-------|--------|---------|-------|----------|--------|--------|----------|------|-----|
| 400 | CBOW/400 | 32.5 | 50.2 | 78.9 | 78.7 | 53.6 | 73.6 | 79.0 | 69.6 | 48.9 | 86.7 |
| | CMOW/400 | **34.4** | 68.8 | 80.1 | **79.9** | 59.8 | 81.9 | **79.2** | 70.7 | **50.3** | 70.7 |
| | H-CBOW | 31.2 | 50.2 | 77.2 | 78.8 | 52.6 | 77.5 | 76.1 | 66.1 | 49.2 | **87.2** |
| | H-CMOW | 32.3 | **70.8** | **81.3** | 76.0 | 59.6 | **82.3** | 77.4 | 70.0 | 50.2 | 38.2 |
| 784 | CBOW/784 | 33.0 | 49.6 | 79.3 | 78.4 | 53.6 | 74.5 | 78.6 | 72.0 | 49.6 | **89.5** |
| 800 | CMOW/784 | **35.1** | 70.8 | **82.0** | 80.2 | **61.8** | 82.8 | **79.7** | 74.2 | **50.7** | 72.9 |
| | Hybrid | 35.0 | **70.8** | 81.7 | **81.0** | 59.4 | **84.4** | 79.0 | **74.3** | 49.3 | 87.6 |
| - | cmp. CBOW | +6.1% | +42.7% | +3% | +3.3% | +10.8% | +13.3% | +0.5% | +3.2% | -0.6% | -2.1% |
| - | cmp. CMOW | -0.3% | +-0% | -0.4% | +1% | -3.9% | +1.9% | -0.9% | +0.1% | -2.8% | +20.9% |

Table 2: Scores on supervised downstream tasks attained by our models. Rows starting with "Cmp." show the relative change with respect to Hybrid.

| Method | SUBJ | CR | MR | MPQA | MRPC | TREC | SICK-E | SST2 | SST5 | STS-B | SICK-R |
|--------|------|-----|-----|------|------|------|--------|------|------|-------|--------|
| CBOW/784 | 90.0 | **79.2** | **74.0** | 87.1 | 71.6 | 85.6 | 78.9 | 78.5 | 42.1 | 61.0 | **78.1** |
| CMOW/784 | 87.5 | 73.4 | 70.6 | **87.3** | 69.6 | **88.0** | 77.2 | 74.7 | 37.9 | 56.5 | 76.2 |
| Hybrid | **90.2** | 78.7 | 73.7 | **87.3** | **72.7** | 87.6 | **79.4** | **79.6** | **43.3** | **63.4** | 77.8 |
| cmp. CBOW | +0.2% | -0.6% | -0.4% | +0.2% | +1.5% | +2.3% | +0.6% | +1.4% | +2.9% | +3.9% | -0.4% |
| cmp. CMOW | +3.1% | +7.2% | +4.4% | +0% | +4.5% | -0.5% | +2.9% | +6.7% | +14.3 | +12.2% | +2.1% |

TREC by up to 4 points. On the TREC task, on the other hand, CMOW outperforms CBOW by 2.5 points.

Our jointly trained model is not more than 0.8 points below the better one of CBOW and CMOW on any of the considered supervised downstream tasks. On 7 out of 11 supervised tasks, the joint model even improves upon the better model, and on SST2, SST5, and MRPC the difference is more than 1 point. The average relative improvement over all tasks is 1.2%.

Regarding the unsupervised downstream tasks (Table 3), CBOW is clearly superior to CMOW on all datasets by wide margins. For example, on STS13, CBOW's score is 50% higher. The hybrid model is able to repair this deficit, reducing the difference to 8%. It even outperforms CBOW on two of the tasks, and yields a slight improvement of 0.5% on average over all unsupervised downstream tasks. However, the variance in relative performance is notably larger than on the supervised downstream tasks.

## 5   DISCUSSION

Our CMOW model produces sentence embeddings that are approximately at the level of fast-Sent (Hill et al., 2016). Thus, CMOW is a reasonable choice as a sentence encoder. Essential to the success of our training schema for the CMOW model are two changes to the original word2vec training. First, our initialization strategy improved the downstream performance by 2.8% compared to Glorot initialization. Secondly, by choosing the target word of the objective at random, the performance of CMOW on downstream tasks improved by 20.8% on average. Hence, our novel training scheme is the first that provides an effective way to obtain parameters for the Compositional Matrix Space Model of language from unlabeled, large-scale datasets.

Table 3: Scores on unsupervised downstream tasks attained by our models. Rows starting with "Cmp." show the relative change with respect to Hybrid.

| Method | STS12 | STS13 | STS14 | STS15 | STS16 |
|--------|-------|-------|-------|-------|-------|
| CBOW | 43.5 | **50.0** | **57.7** | **63.2** | 61.0 |
| CMOW | 39.2 | 31.9 | 38.7 | 49.7 | 52.2 |
| Hybrid | **49.6** | 46.0 | 55.1 | 62.4 | **62.1** |
| cmp. CBOW | +14.6% | -8% | -4.5% | -1.5% | +1.8% |
| cmp. CMOW | +26.5% | +44.2% | +42.4 | +25.6% | +19.0% |

Regarding the probing tasks, we observe that CMOW embeddings better encode the linguistic properties of sentences than CBOW. CMOW gets reasonably close to CBOW on some downstream tasks. However, CMOW does not in general supersede CBOW embeddings. This can be explained by the fact that CBOW is stronger at word content memorization, which is known to highly correlate with the performance on most downstream tasks (Conneau et al., 2018). Yet, CMOW has an increased performance on the TREC question type classification task (88.0 compared to 85.6). The rationale is that this particular TREC task belongs to a class of downstream tasks that require capturing other linguistic properties apart from Word Content (Conneau et al., 2018).

Due to joint training, our hybrid model learns to pick up the best features from CBOW and CMOW simultaneously. It enables both models to focus on their respective strengths. This can best be seen by observing that H-CMOW almost completely loses its ability to memorize word content. In return, H-CMOW has more capacity to learn other properties, as seen in the increase in performance at BShift and others. A complementary behavior can be observed for H-CBOW, whose scores on Word Content are increased. Consequently, with an 8% improvement on average, the hybrid model is substantially more linguistically informed than CBOW. This transfers to an overall performance improvement by 1.2% on average over 11 supervised downstream tasks, with large improvements on sentiment analysis tasks (SST2, SST5), question classification (TREC), and the sentence representation benchmark (STS-B). The improvements on these tasks is expected because they arguably depend on word order information. On the other tasks, the differences are small. Again, this can be explained by the fact that most tasks in the SentEval framework mainly depend on word content memorization (Conneau et al., 2018), where the hybrid model does not improve upon CBOW.

Please note, the models in our study do not represent the state-of-the-art for sentence embeddings. Perone et al. (2018) show that better scores are achieved by LSTMs and Transformer models, but also by averaging word embedding from fastText (Mikolov et al., 2018). These embeddings were trained on the CBOW objective, and are thus very similar to our models. However, they are trained on large corpora (600B tokens vs 3B in our study), use large vocabularies (2M vs 30k in our study), and incorporate numerous tricks to further enhance the quality of their models: word subsampling, subword-information, phrase representation, n-gram representations, position-dependent weighting, and corpus de-duplication. In the present study, we focus on comparing CBOW, CMOW, and the hybrid model in a scenario where we have full control over the independent variables. To single out the effect of the independent variables better, we keep our models relatively simple. Our analysis yields interesting insights on what our models learn when trained separately or jointly, which we consider more valuable in the long term for the research field of text representation learning.

We offer an efficient order-aware extension to embedding algorithms from the bag-of-words family. Our 784-dimensional CMOW embeddings can be computed at the same rate as CBOW embeddings. We empirically measured in our experiments 71k for CMOW vs. 61k for CBOW in terms of encoding sentences per second. This is because of the fast implementation of matrix multiplication in GPUs. It allows us to encode sentences approximately 5 times faster than using a simple Elman RNN of the same size (12k per second). Our matrix embedding approach also offers valuable theoretical advantages over RNNs and other autoregressive models. Matrix multiplication is associative such that only $\log_2 n$ sequential steps are necessary to encode a sequence of size $n$. Besides parallelization, also dynamic programming techniques can be employed to further reduce the number of matrix multiplication steps, e. g., by pre-computing frequent bigrams. We therefore expect our matrix embedding approach to be specifically well-suited for large-scale, time-sensitive text encoding applications. Our hybrid model serves as a blueprint for using CMOW in conjunction with other existing embedding techniques such as fastText (Mikolov et al., 2018).

## 6    CONCLUSION

We have presented the first efficient, unsupervised learning scheme for the word order aware Compositional Matrix Space Model. We showed that the resulting sentence embeddings capture linguistic features that are complementary to CBOW embeddings. We thereupon presented a hybrid model with CBOW that is able to combine the complementary strengths of both models to yield an improved downstream task performance, in particular on tasks that depend on word order information. Thus, our model narrows the gap in terms of representational power between simple word embedding based sentence encoders and highly non-linear recurrent sentence encoders.

We made the code for this paper available at `https://github.com/florianmai/word2mat`.

ACKNOWLEDGEMENT

This research was supported by the Swiss National Science Foundation under the project Learning Representations of Abstraction for Opinion Summarisation (LAOS), grant number "FNS-30216".

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

APPENDICES

## A  PROOF OF CONSTANT EXPECTED VALUE OF MATRIX MULTIPLICATION

The statement that we formally proof is the following. For any sequence $s = s_1 \ldots s_n$:

$$\forall 1 \leq k \leq n : \mathbb{E}[\text{enc}_{\prod}(s_1, \ldots, s_k)] = \boldsymbol{I}_d.$$

The basis ($n = 1$) follows trivially due to the expected value of each entry being the mean of the normal distribution. For the induction step, let $\mathbb{E}[\prod_{i=1}^{n}(W_i)] = \boldsymbol{I}_d$. It follows:

$$\mathbb{E}[\prod_{i=1}^{n+1}(W_i)]$$

$$= \mathbb{E}[\prod_{i=1}^{n}(W_i) \cdot W_{n+1}]$$

$$= \mathbb{E}[\prod_{i=1}^{n}(W_i)] \cdot \mathbb{E}[W_{n+1}] \qquad \text{(Independence)}$$

$$= \boldsymbol{I}_d \cdot \mathbb{E}[W_{n+1}] \qquad \text{(Hypothesis)}$$

$$= \boldsymbol{I}_d \cdot \boldsymbol{I}_d \qquad \text{(Exp. val of each entry)}$$

$$= \boldsymbol{I}_d$$

## B  FURTHER EXPERIMENTS AND RESULTS

### B.1  COMPARISON OF OBJECTIVES

In Section 3.2, we describe a more general training objective than the classical CBOW objective from Mikolov et al. (2013a). The original objective always sets the center word from the window of tokens $(w_{t-c}, \ldots, w_{t+c})$ as target word, $w_O = w_t$. In preliminary experiments, this did not yield satisfactory results. We believe that this objective is too simple for learning sentence embeddings that capture semantic information. Therefore, we experimented a variant where the target word is sampled randomly from a uniform distribution, $w_O := \mathcal{U}(\{w_{t-c}, \ldots, w_{t+c}\})$.

To test the effectiveness of this modified objective, we evaluate it with the same experimental setup as described in Section 4. Table 4 lists the results on the linguistic probing tasks. CMOW-C and CBOW-C refer to the models where the center word is used as the target. CMOW-R and CBOW-R refer to the models where the target word is sampled randomly. While CMOW-R and CMOW-C perform comparably on most probing tasks, CMOW-C yields 5 points lower scores on WordContent and BigramShift. Consequently, CMOW-R also outperforms CMOW-C on 10 out of 11 supervised downstream tasks and on all unsupervised downstream tasks, as shown in Tables 5 and 6, respectively. On average over all downstream tasks, the relative improvement is 20.8%. For CBOW, the scores on downstream tasks increase on some tasks and decrease on others. The differences are miniscule. On average over all 16 downstream tasks, CBOW-R scores 0.1% lower than CBOW-C.

Table 4: Scores for different training objectives on the linguistic probing tasks.

| Method | Depth | BShift | SubjNum | Tense | CoordInv | Length | ObjNum | TopConst | SOMO | WC |
|--------|-------|--------|---------|-------|----------|--------|--------|----------|------|-----|
| CMOW-C | **36.2** | 66.0 | 81.1 | 78.7 | 61.7 | **83.9** | 79.1 | 73.6 | 50.4 | 66.8 |
| CMOW-R | 35.1 | **70.8** | **82.0** | **80.2** | **61.8** | 82.8 | **79.7** | **74.2** | **50.7** | **72.9** |
| CBOW-C | **34.3** | **50.5** | **79.8** | **79.9** | 53.0 | **75.9** | **79.8** | **72.9** | 48.6 | 89.0 |
| CBOW-R | 33.0 | 49.6 | 79.3 | 78.4 | **53.6** | 74.5 | 78.6 | 72.0 | **49.6** | **89.5** |

Table 5: Scores for different training objectives on the supervised downstream tasks.

| Method | SUBJ | CR | MR | MPQA | MRPC | TREC | SICK-E | SST2 | SST5 | STS-B | SICK-R |
|--------|------|------|------|------|------|------|--------|------|------|-------|--------|
| CMOW-C | 85.9 | 72.1 | 69.4 | 87.0 | **71.9** | 85.4 | 74.2 | 73.8 | 37.6 | 54.6 | 71.3 |
| CMOW-R | **87.5** | **73.4** | **70.6** | **87.3** | 69.6 | **88.0** | **77.2** | **74.7** | **37.9** | **56.5** | **76.2** |
| CBOW-C | **90.0** | **79.3** | **74.6** | **87.5** | **72.9** | 85.0 | **80.0** | 78.4 | 41.0 | 60.5 | **79.2** |
| CBOW-R | **90.0** | 79.2 | 74.0 | 87.1 | 71.6 | **85.6** | 78.9 | **78.5** | **42.1** | **61.0** | 78.1 |

Table 6: Scores for different training objectives on the unsupervised downstream tasks.

| Method | STS12 | STS13 | STS14 | STS15 | STS16 |
|--------|-------|-------|-------|-------|-------|
| CMOW-C | 27.6 | 14.6 | 22.1 | 33.2 | 41.6 |
| CMOW-R | **39.2** | **31.9** | **38.7** | **49.7** | **52.2** |
| CBOW-C | **43.5** | 49.2 | **57.9** | **63.7** | **61.6** |
| CBOW-R | **43.5** | **50.0** | 57.7 | 63.2 | 61.0 |

## B.2 INITIALIZATION STRATEGY

In Section 3.3, we present a novel random initialization strategy. We argue why it is more adequate for training CMSMs than classic strategies that initialize all parameters with random values close to zero, and use it in our experiments to train CMOW.

To verify the effectiveness of our initialization strategy empirically, we evaluate it with the same experimental setup as described in Section 4. The only difference is the initialization strategy, where we include Glorot initialization (Glorot & Bengio, 2010) and the standard initialization from $\mathcal{N}(0, 0.1)$. Table 7 shows the results on the probing tasks. While Glorot achieves slightly better results on BShift and TopConst, CMOW's ability to memorize word content is improved by a wide margin by our initialization strategy. This again affects the downstream performance as shown in Table 8 and 9, respectively: 7 out of 11 supervised downstream tasks and 4 out of 5 unsupervised downstream tasks improve. On average, the relative improvement of our strategy compared to Glorot initialization is 2.8%.

Table 7: Scores for initialization strategies on probing tasks.

| Initialization | Depth | BShift | SubjNum | Tense | CoordInv | Length | ObjNum | TopConst | SOMO | WC |
|----------------|-------|--------|---------|-------|----------|--------|--------|----------|------|------|
| $\mathcal{N}(0, 0.1)$ | 29.7 | 71.5 | 82.0 | 78.5 | 60.1 | 80.5 | 76.3 | 74.7 | **51.3** | 52.5 |
| Glorot | 31.3 | **72.3** | 81.8 | 78.7 | 59.4 | 81.3 | 76.6 | **74.6** | 50.4 | 57.0 |
| Our paper | **35.1** | 70.8 | **82.0** | **80.2** | **61.8** | **82.8** | **79.7** | 74.2 | 50.7 | **72.9** |

Table 8: Scores for initialization strategies on supervised downstream tasks.

| Initialization | SUBJ | CR | MR | MPQA | MRPC | TREC | SICK-E | SST2 | SST5 | STS-B | SICK-R |
|---|---|---|---|---|---|---|---|---|---|---|---|
| $\mathcal{N}(0, 0.1)$ | 85.6 | 71.5 | 68.4 | 86.2 | **71.6** | 86.4 | 73.7 | 72.3 | **38.2** | 53.7 | 72.7 |
| Glorot | 86.2 | **74.4** | 69.5 | 86.5 | 71.4 | **88.4** | 75.4 | 73.2 | **38.2** | 54.1 | 73.6 |
| Our paper | **87.5** | 73.4 | **70.6** | **87.3** | 69.6 | 88.0 | **77.2** | **74.7** | 37.9 | **56.5** | **76.2** |

Table 9: Scores for initialization strategies on unsupervised downstream tasks.

| Initialization | STS12 | STS13 | STS14 | STS15 | STS16 |
|---|---|---|---|---|---|
| $\mathcal{N}(0, 0.1)$ | 37.7 | 26.5 | 33.3 | 44.7 | 50.3 |
| Glorot | **39.6** | 27.2 | 35.2 | 46.5 | 51.6 |
| Our paper | 39.2 | **31.9** | **38.7** | **49.7** | **52.2** |

