# OpenReview forum: "CBOW Is Not All You Need: Combining CBOW with the Compositional Matrix Space Model"
_ICLR.cc/2019/Conference_

### Official Review · AnonReviewer1 · 2018-10-26
**A way to actually train sequential embedding models**

**Rating:** 6
**Confidence:** 3

**Review:**

The main contribution of this paper in practice seems to be a way to initialize the Continuous Matrix Space Model so that training actually converges, followed by a slightly different contrastive loss function used to train these models. The paper explores the pure matrix model and a mixed matrix / vector model, showing that both together improve on simpler methods on many benchmark tasks.

My main concern is that the chained matrix multiplication involved in this method is not substantially simpler than an RNN or LSTM sentence encoding model, and there are no comparisons of training and inference cost between the models proposed in this paper and conceptually simpler RNNs and LSTMs. The FastSent paper, used here as a baseline, does compare against some deep models, but they choose far more complex baselines such as the NMT encoding, which is trained on a very different loss function. Indeed the models proposed here do not seem to outperform fasttext and fastsent despite having fairly similar computational costs.

I think this paper could use a little more justification for when it's appropriate to use the method proposed here versus more straightforward baselines.

---

> ### Author Response · Authors · 2018-11-19
> **Our model is more efficient than RNNs**
>
> Dear reviewer,
>
> Thank you for your helpful comments. From my understanding, your main critique is:
>
> 1. “No comparison in terms of encoding speed with RNNs”
> 2. “Our model does not yield as good performance as fastText and fastSent”
>
> Let me respond to these points one by one.
>
> 1.
> We did not do compare against RNNs in terms of encoding speed, because it is already clear from the corpus of related work that embeddings from RNNs are much slower to compute than CBOW embeddings (see for instance Hill et al. (2016), to which you also refer).  Since we report that CMOW is approximately as fast as CBOW, we thought that this is clear, i.e., that our method is much more efficient than RNNs.
>
> However, it seems that this is not clear from our paper. Thus, we performed some measurements of our own and added the results to the paper (last paragraph of Discussion section). We had already reported the encoding speeds of CBOW and CMOW at test time (61k and 71k sentences per second, respectively). We also added the results of an Elman RNN in order to show that CMOW is substantially faster: In our experiments, the Elman RNN encodes 12k sentences per second, which is 5 times slower than our CMOW encoder. This corresponds almost exactly to the results also observed by Hill et al. at test time.
>
> Please note, CMOW and CBOW are based on matrix multiplication and addition, respectively, which are associative operations. As such, CMOW and CBOW have substantial parallelization capacities: For a sequence of length n, only log(n) sequential steps are required, and the rest can be computed in parallel. On the other hand, an RNN is not associative and cannot be parallelized in the same manner: It requires n sequential steps.
>
> 2.
> First, I would like to point out that our Hybrid method DOES outperform the results from fastSent on all supervised datasets that they report results on. On TREC and MRPC, the differences are even as large as 14.1% and 8.3% improvement, respectively.
>
> However, the training settings are too different to be considered fair, which we point out in the paper repeatedly: FastSent is trained on a corpus that is three times smaller (1B tokens vs 3B tokens in our case).
> It is important to understand that we do not consider fastSent as a baseline to directly compare to. We report their results merely to show that our methods (which perform better than fastSent) produce embeddings that are reasonably useful.
> The same holds for fastText: This was trained on a much larger corpus than our methods (600B tokens as opposed to 3B tokens) and its vocabulary has 2M words as opposed to 30k in our case. Hence, again, this comparison is by no means fair.
> We perform a controlled study which allows us to identify exactly where the differences in performance come from. Hence, the only direct baseline is the CBOW model from our paper.
>
> Hill et al. (2016) : Learning Distributed Representations of Sentences from Unlabelled Data, NAACL 2016

---

> > ### Comment · AnonReviewer1 · 2018-11-19
> > **response to author response**
> >
> > After reading the author response I'm revising my scores upward.

---

### Official Review · AnonReviewer2 · 2018-11-03
**Interesting model to embed words in a way that captures order information**

**Rating:** 5
**Confidence:** 4

**Review:**


The authors propose CMOW, an extension of the CBOW model that allows the model to capture word order. Instead of each word being represented as a vector, words are represented by matrices. They extend the CBOW objective to take into account word order by replacing the averaging of vectors to create the context with matrix multiplication (a non-commutative operation). This is the first time this model has been applied in a large scale unsupervised setting. They are able to do this using their objective and an initialization strategy where the matrix embeddings are set to the identity matrix with some Gaussian noise added.

The results of this paper are its main weakness. I did enjoy reading the paper, and it is nice to see some results using matrices as embeddings and matrix multiplication as a compositional function. They include a nice analysis of how word order is captured by these CMOW embeddings while CBOW embeddings capture the word content, but it doesn't seem to make much of a difference on the downstream tasks where CBOW is better than CMOW and close to the performance of the hybrid combination of CBOW and CMOW.

I think it's clear that their model is able to capture word information to some extent, but other models  (RNNs etc.) can do this as well, that admittedly are more expensive, but also have better performance on downstream tasks. I think a stronger motivation for their method besides an analysis of some phenomena it captures and a slight improvement on some downstream tasks when combined with CBOW is needed though for acceptance. Could it be used in other settings besides these downstream transfer tasks?

PROS:
- introduced an efficient and stable approach for training CMSM models
- Show that their model CMOW is able to capture word order information
- Show that CMOW compliments CBOW and a hybrid model leads to improved results on downstream tasks.

CONS
- The results on the hybrid model are only slightly better than CBOW. CMOW alone is mostly worse than CBOW.

---

> ### Author Response · Authors · 2018-11-19
> **Enhancing simple word embedding models is an common, important topic of research**
>
> Dear reviewer,
>
> Thanks for your review! To my understanding, your main concerns with our paper are:
>
> 1. “The improvements of the Hybrid model over CBOW are not large enough”
> 2. “There are other, more powerful models (RNNs) that achieve much better results, so there is not enough justification.”
>
> Let me address these issues one at a time:
>
> 1.
> We conducted a study in the field of learning universal sentence embeddings. Obviously, an embedding that doesn’t have a notion of word order should not be considered "universal". The goal of our research is thus to push the limits of what simple word aggregation methods are capable of encoding. Finding some empirical evidence, Henao et al. (2018) hypothesize that the main difference of simple word embedding methods to RNNs may be their inability to capture word order.
>
> We successfully propose a way to diminish that difference. Our hybrid CBOW-CMOW model is not only able to capture word order information, it scores 8% better on average on the linguistic probing tasks than CBOW. Even if we disregard the benefit from BShift, the improvement is still large (~4%). From the perspective of learning linguistically informed universal sentence embeddings, this is an important result, especially at a conference that is all about learning representations.
>
> It is true that the results on linguistic probing tasks do not transfer to the same extent to the downstream tasks, achieving an average improvement of "only" 1.2%. We have added this in the revised version of the paper.
> We evaluate our models on the SentEval benchmark. This framework is the de facto standard for evaluating sentence embeddings, and thus we should evaluate our models this way as well. Most tasks in SentEval depend heavily on word content memorization (Conneau et al., 2018). Thus, the selection of downstream tasks rather disfavors our model, since it improves in every aspect but Word Content memorization.
> Recently, more doubt has been cast repeatedly whether the selection of tasks in SentEval is sufficient to test the generality of sentence embeddings (“Anonymous ICRL Submission”, 2018), especially their compositionality (Dasgupta et al., 2018).
>
> In summary, considering the strong results on linguistic probing tasks, and the nature of the SentEval framework, we believe that the results obtained by our hybrid CBOW-CMOW model are already strong evidence that our method produces more general, robust sentence embeddings.
>
> 2.
>
> The research community in sentence embedding learning has paid a lot of attention to baselines based on word embedding aggregation methods (such as the one presented in this paper) that are conceptually simple, e.g., Henao et al. (2018), Pagliardini et al. (2018), Rueckle et al. (2018), including important work presented at ICLR (Wieting et al (2016), Arora et al. (2017).
> The reasoning is two-fold: i) Aggregated word embeddings are computationally inexpensive compared to RNNs (see Hill et al. (2016), and the measurements in our work). ii) Pushing the limits of conceptually simple encoders helps to identify the benefit introduced by more sophisticated encoders, which has also been a recurring topic of interest ( Adi et al. (2016), Conneau et al. (2018), Zhu et al. (2018), Anonymous (2018) ).
> Our paper is clearly motivated by reason i), since our method is computationally as inexpensive as CBOW. It is also motivated by reason ii): The conceptual difference between CBOW and CMOW boils down to using matrix multiplication instead of addition, followed by simple adaptations to the training procedure. Yet, these changes substantially improve the model's ability to learn linguistic properties such as word order, which were formerly left up to more sophisticated RNNs.
>
> Adi et al. (2016) : Fine-grained Analysis of Sentence Embeddings Using Auxiliary Prediction Tasks, ICLR 2017
> Anonymous (2018) : No Training Required: Exploring Random Encoders for Sentence Classification. URL: https://openreview.net/forum?id=BkgPajAcY7 , ICLR 2019 Submission
> Arora et al. (2017) : A Simple But Tough-to-Beat Baseline for Sentence Embeddings, ICLR 2017
> Conneau et al. (2018): What you can cram into a single vector, ACL 2018
> Henao et al. (2018) : Baseline Needs More Love: On Simple Word-Embedding-Based Models and Associated Pooling Mechanisms, ACL 2018
> Hill et al. (2016) : Learning Distributed Representations of Sentences from Unlabelled Data, NAACL 2016
> Pagliardini et al. (2018) : Unsupervised Learning of Sentence Embeddings using Compositional n-Gram Features, NAACL 2018
> Rueckle et al. (2018) : Concatenated Power Mean Word Embeddings as Universal Cross-Lingual Sentence Representations, arXiv:1803.01400
> Wieting et al. (2016) : Towards Universal Paraphrastic Sentence Embeddings, ICLR 2016
> Zhu et al. (2018) : Exploring Semantic Properties of Sentence Embeddings

---

### Official Review · AnonReviewer3 · 2018-11-03
**new training schemes for a matrix-multiplicative variant of CBOW**

**Rating:** 6
**Confidence:** 4

**Review:**

The paper presents new training schemes and experiments for a matrix-multiplicative variant of CBOW. This variant is called a CMSM (Yessenalina and Cardie, 2011; Asaadi and Rudolph, 2017) which swaps the bag of vectors to a product of square matrices for encoding context to incorporate word ordering. It seems this model has not been trained successfully before (at least with a simple approach) due to the vanishing gradient problem.

The paper's main contributions are an initialization scheme for context matrices (to I + [N(0,0.1)]) to counter the vanishing gradient problem and a modification of the CBOW objective so that the target word is drawn uniformly at random from the context window (rather than the center word). Both are shown to improve the quality of learned representations when evaluated as sentence embeddings. Concatenating CBOW and CMSM architectures is additionally helpful.

I was not aware of the matrix-multiplicative variant of CBOW previously so it's possible that I don't have the expertise to judge the novelty of the approach. But the idea is certainly sensible and the proposed strategies seem to work. The main downside is that for all this work the improvements seem a little weak. The averaged fastText embeddings are clearly superior across the board, though as the authors say it's probably unfair to compare based on different training settings. But this doesn't hurt the simplicity and effectiveness of the proposed method when compared against CBOW baselines.

---

> ### Author Response · Authors · 2018-11-19
> **The hybrid CBOW-CMOW model makes CBOW a more robust baseline**
>
> Dear reviewer,
>
> Thank you for your comments! To my understanding, your main concerns with our paper are the following:
>
> 1. “fastText embeddings are clearly better than our approach.”
> 2. “The improvements of the Hybrid model over CBOW are small”.
>
> I would like to address these concerns in the following.
>
> 1.
> We aim at conducting a controlled study, where we have full control over the independent variables. This allows me to precisely measure the effect of our changes/extensions to the CBOW model. Therefore, our baseline is the CBOW we trained ourselves, not fastText or fastSent. We report the scores of fastText and fastSent merely to show that our models produce useful embeddings and are therefore worth studying in the first place. FastText and FastSent are NOT the baselines we compare against.
>
> Let me elaborate why the scores we report for fastText are not comparable to our approach. The scores achieved by fastText are based on the implementation by Mikolov et al. (2018). Like our baseline, fastText is based on the CBOW objective, i.e., predicting the center word from the sum of its context word embeddings. However, their model is trained on a much larger corpus (CommonCrawl, 630B tokens, vs UMBC, 3B tokens), and with a much larger vocabulary (2M words vs. 30,000 in our case).
>
> Furthermore, the authors of fastText employ many tricks to enhance the quality of their models (word subsampling, subword-information, phrase representation, n-gram representations, etc.). For simplicity, I focus on the essential part of our models, i.e., the composition function, in order to conduct a fair and scientifically robust comparison of the performance of CBOW with my novel CMOW and finally the hybrid CBOW-CBOW-model. This makes a direct comparison with fastText very difficult, if not entirely unfair.
>
> 2.
> My paper is concerned with learning universal sentence embeddings with simple word embedding methods. Averaging word embeddings already shows good performance on downstream tasks. However, one cannot really expect to obtain a "universal" sentence embedding from an encoder that is word-order agnostic like CBOW. In fact, finding some empirical evidence, Henao et al. (2018) recently hypothesized that word-order sensitivity may be the main difference of simple word aggregation methods to RNNs.
> We successfully propose a method to diminish this difference.
> Our hybrid CBOW-CMOW-model is not only able to capture word order information like RNNs. It also scores on average 8% better on the linguistic probing tasks than CBOW! Even if we disregard the benefit from BShift, the improvement is still large (~4%). From the perspective of learning linguistically informed universal sentence embeddings, this is an important result.
>
> It is true that the results on linguistic probing tasks do not transfer to the same extent to the downstream tasks, achieving an average improvement of "only" 1.2%. We have added this in the revised version of the paper.
> We evaluate our models on the SentEval benchmark. This framework is the de facto standard for evaluating sentence embeddings, and thus we should evaluate our models this way as well. Most tasks in SentEval depend heavily on word content memorization (Conneau et al., 2018). Thus, the selection of downstream tasks rather disfavors our model, since it improves in every aspect but Word Content memorization.
> Recently, more doubt has been cast repeatedly whether the selection of tasks in SentEval is sufficient to test the generality of sentence embeddings (“Anonymous ICRL Submission”, 2018), especially their compositionality (Dasgupta et al., 2018).
>
> In summary, considering the strong results on linguistic probing tasks, and the nature of the SentEval framework, we believe that the results obtained by our hybrid CBOW-CMOW model are already strong evidence that our method produces more general, robust sentence embeddings.
>
> “Anonymous ICRL Submission”(2018): No Training Required: Exploring Random Encoders for Sentence Classification. URL: https://openreview.net/forum?id=BkgPajAcY7
> Conneau et al. (2018): What you can cram into a single vector, ACL 2018
> Dasgupta et al. (2018): Evaluating Compositionality in Sentence Embeddings, arXiv:1802.04302
> Henao et al. (2018) : Baseline Needs More Love: On Simple Word-Embedding-Based Models and Associated Pooling Mechanisms, ACL 2018
> Mikolov et al. (2018): Advances in Pre-Training Distributed Word Representations, LREC 2018

---

> ### Comment · AnonReviewer3 · 2018-11-24
> **post-response note**
>
> I understand the intent/value of doing controlled experiments. While I said the main weakness was weak experimental results, now I think a bigger issue is the impact of the work on the broader ICLR community. It is indeed a rather specialized contribution about a specific problem and technique, so while I like the paper I'm a bit hesitant to advocate it more strongly. Hence I'll keep my rating.

---

> > ### Author Response · Authors · 2018-12-01
> > **post-response note response**
> >
> > Dear Reviewer,
> >
> > Due to its brevity, your comment leaves a lot of room for interpretation, and we are not sure if we understood your concerns correctly. Nevertheless, we would like to address them in the following.
> >
> > To our understanding, your main concerns with our paper now is that
> >
> > 1. We study a problem that is too “specific”, i.e., not of interest for a broad audience at ICLR.
> > 2. Our contribution is too “specialized”, i.e., not of large value for the considered problem.
> > 3. The technique is too “specific”, i.e., not well studied and thus does not integrate well with common approaches.
> >
> > We strongly disagree with you on all three of these points.
> >
> > 1.
> > Sentence representation learning has been widely studied in recent years, mainly because it has many important applications. In our paper alone we consider more than 10 supervised downstream tasks and several unsupervised downstream tasks.
> > The importance of this topic for ICLR can also be seen from the fact that more than a dozen submissions have “Sentence embedding”, “sentence representation”, or “sentence encoding” in their title alone!
> > Obviously, being concerned with unsupervised representation learning for NLP, our paper is also very well in line with ICLR 2019’s CfP.
> >
> > 2.
> > Our contribution is to introduce order awareness to efficient, linear word embedding based models. Given that semantics of natural language are inherently order dependent, this has considerable consequences for the expressiveness of the resulting model.
> > Hence, our paper can not be considered a “specialized” contribution.
> >
> > 3.
> > CMOW is based on the Compositional Matrix Space Model, which is a rather old idea that indeed has not been studied much. However, due to its conceptual similarity to CBOW, some knowledge is transferable, such that it can be trained in a similar fashion as CBOW.
> >
> > Our hybrid CBOW-CMOW model extends and improves upon CBOW, which is arguably the most popular baseline for sentence representation. It consists of a simple concatenation of CBOW and CMOW at the embedding level. As such, CMOW can be integrated with existing CBOW approaches easily.
> >
> > To get a word order aware text embedding model, the obvious choice would have been some kind of RNN. This would probably be considered less “specialized” or “specific”. But keep in mind that science demands not only pushing the limits of already well-established techniques, but also following underexplored research paths. Considering the reviewer guidelines, https://iclr.cc/Conferences/2019/Reviewer_Guidelines , ICLR seems to be a place that welcomes these efforts.
> >
> > Although we would appreciate a higher rating, we would like to thank you for advocating our paper!

---

### Meta-Review · Area_Chair1 · 2018-12-14
**Clear study of an important problem, though improvements limited**

**Confidence:** 4
**Recommendation:** Accept (Poster)

**Metareview:**

This paper presents CMOW—an unsupervised sentence representation learning method that treats sentences as the product of their word matrices. This method is not entirely novel, as the authors acknowledge, but it has not been successfully applied to downstream tasks before. This paper presents methods for successfully training it, and shows results on the SentEval benchmark suite for sentence representations and an associated set of analysis tasks.

All three reviewers agree that the results are unimpressive: CMOW is no better than the faster CBOW baseline on most tasks, and the combination of the two is only marginally better than CBOW. However, CMOW does show some real advantages on the analysis tasks. No reviewer has any major correctness concerns that I can see.

As I see it, this paper is borderline, but narrowly worth accepting: As a methods paper, it presents weak results, and it's not likely that many practitioners will leap to use the method. However, the method is so appealingly simple and well known that there is some value in seeing this as an analysis paper that thoroughly evaluates it. Because it is so simple, it will likely be of interest to researchers beyond just the NLP domain in which it is tested (as CBOW-style models have been), so ICLR seems like an appropriate venue. It seems like it's in the community's best interest to see a method like this be evaluated, and since this paper appears to offer a thorough and sound evaluation, I recommend acceptance.